# Effects of an aquatic protocol on electromyography activation and strength of lower limb muscles in blind women: A randomized controlled trial

Asma Salari[1], Mansour Sahebozamani[2], Abdolhamid Daneshjoo[2], Mohammad Alimoradi[2], Mojtaba Iranmanesh[2], Nicola Relph [3]*, Guillermo Mendez-Rebolledo [4,5]

**1** Department of Sports Sciences and Physical Education, Faculty of Humanities Science, Gonbad Kavoos University, Gonbad Kavoos, Iran, **2** Department of Sports Injuries and Corrective Exercises, Faculty of Sports Sciences, University of Shahid Bahonar, Kerman, Iran, **3** *Faculty of Health, Social Care and Medicine, Edge Hill University, Ormskirk, United Kingdom, **4** Laboratorio de Investigación Somatosensorial y Motora, Escuela de Kinesiología, Facultad de Salud, Universidad Santo Tomás, Talca, Chile, **5** Magíster en Ciencias de la Actividad Física y del Deporte Aplicadas al Entrenamiento, Rehabilitación y Reintegro Deportivo, Escuela de Kinesiología, Facultad de Salud, Universidad Santo Tomás, Santiago, Chile

* Nicola.Relph@edgehill.ac.uk

## Abstract

### Purpose

Visual impairment poses considerable challenges to mobility and everyday tasks, frequently leading to a more sedentary lifestyle and reduced physical fitness levels. Therefore, this study investigated the effects of a tailored aquatic exercise protocol on muscle activation and strength in visually impaired individuals.

### Materials and methods

Thirty women who were blind (mean age = 29.03 ± 2.20 years) were randomly assigned to an experimental (EX) group and a control (CO) group. The EX group participated in three weekly 60-minute aquatic sessions, while the CO group maintained regular activities. Electromyography (EMG) activation and onset time were measured in the tibialis anterior, gastrocnemius medialis, rectus femoris, and biceps femoris. Muscle strength was also assessed in the ankle dorsiflexors and plantarflexors, as well as the knee flexors and extensors.

### Results

The EX group showed increased EMG activation for the tibialis anterior, gastrocnemius medialis, rectus femoris, and biceps femoris in both anterior-posterior (2.23 MVIC%, 95% CI: 1.13 to 3.34, p < 0.001) and posterior-anterior directions (2.35 MVIC%, 95% CI: 1.80 to 2.91, p < 0.001) compared to CO group.
Onset time decreased significantly in the EX group relative to CO group

**Data availability statement:** The data that supports the findings of this study are available in the Zenodo data repository at https://doi.org/10.5281/zenodo.14280848.

**Funding:** The author(s) received no specific funding for this work.

**Competing interests:** The authors have declared that no competing interests exist.

(anterior-posterior: -108.07 ms, 95% CI: -117.23 to -98.89, p < 0.001; posterior-anterior: -98.72 ms, 95% CI: -106.54 to -90.90, p < 0.001). Muscle strength significantly increased in the EX group compared to the CO group, with greater strength in ankle dorsiflexors (4.42 N/kg, 95% CI: 3.42 to 5.42, p < 0.001) and knee extensors (2.72 N/kg, 95% CI: 1.04 to 4.40, p < 0.001).

## Conclusions

The aquatic exercise program improved neuromuscular function and strength in women with visual impairments, supporting its use in rehabilitation.

## Trial registration

IRCT2017022132705N1

---

## Introduction

Visual impairment significantly affects daily functioning for millions worldwide, with a disproportionate impact in low-income countries [1–3]. It can lead to social isolation, decreased mobility, and reduced physical activity, contributing to diminished physical fitness [4,5]. Additionally, individuals with visual impairments often experience deficits in postural control, muscle weakness, and impaired balance, heightening the risk of falls with potentially severe consequences [6–8]. Muscle activation and strength play a crucial role in maintaining stability and functional ability, directly impacting postural balance [9,10]. Research indicates that lower limb muscle weakness is associated with greater center of pressure sways, leading to impaired postural control and increased fall risk [11,12].

Previous research has demonstrated positive outcomes from interventions aimed at enhancing postural control in individuals with visual impairments [13,14]. Among these, aquatic exercises have shown promise due to their ability to provide resistance, promoting muscle engagement and strength development [15]. The buoyancy and hydrostatic pressure in water offer support and stability while challenging muscles to enhance strength, flexibility, and balance [16–18]. Although the negative impact of visual impairment on postural control is well established, the effectiveness of aquatic exercises in improving neuromuscular function remains less understood.

Postural control relies on three systems: visual, vestibular, and somatosensory. A deficiency in one system necessitates compensation by the other two [14]. Therefore, interventions must harness the vestibular and somatosensory systems to enhance balance [19]. Studies have explored exercise programs targeting balance deficits in individuals with visual impairments [20,21]. For example, di Cagno et al. (2018) found no impairment in static balance after whole-body vibration in congenitally blind men [22]. Similarly, Salari et al. (2024) reported that regular aquatic exercise programs improved postural stability in visually impaired women. The buoyant environment of water reduces fear of falling and allows greater movement freedom, making aquatic exercises effective for balance training [21].

Targeting sensorimotor deficits through such interventions could reduce fall risks and enhance functional outcomes. This study aims to explore the effects of an aquatic exercise protocol on electromyography (EMG) activation and lower limb muscle strength in women with visual impairments. We hypothesize that this protocol will improve EMG activation and muscle strength, ultimately enhancing postural control and reducing fall risks. Understanding these potential benefits could inform targeted rehabilitation strategies to improve functional ability and quality of life in individuals with visual impairments.

## Materials and methods

### Design

For the present study, a single-blind Randomized Controlled Trial (RCT) design was employed following CONSORT reporting guidelines (see supporting information [S1 file CONSORT-2010-Checklist]). The study was registered in the Iranian Registry of Clinical Trials (IRCT2017022132705N1). Full protocols are available in the supporting information files [S2 file Research Protocol - English] and [S3 file Research Protocol – Persian].

The sample size was determined using Equation 1, incorporating parameters derived from prior studies and statistical power considerations. The calculation was conducted with a confidence level of 95% ($Z_{1-\alpha/2} = 1.96$) and a statistical power of 80% ($Z_{1-\beta} = 0.80$), with an effect size of 0.46 based on similar aquatic exercise intervention in visually impaired individuals [21].

$$N = \frac{\left(Z_{1-\frac{a}{2}} + Z_{1-\beta}\right)^2 \left(S_1^2 + S_2^2\right)^2}{(M_1 - M_2)2} \tag{1}$$

where $Z_{1-\alpha/2}$ represents the critical value for the chosen confidence level, $Z_{1-\beta}$ represents the critical value for statistical power, $S_1$ and $S_2$ are the standard deviations of the two groups, obtained from prior literature, and $M_1$ and $M_2$ are the expected means based on previous studies. Using this formula, we determined that a minimum of 12 participants per group was necessary. To account for potential dropouts, we increased the sample size to 15 participants per group (n = 30 total), ensuring sufficient statistical power. Cohen's d formula was determined using Equation 2 to compute this effect size, ensuring its relevance to our study population [23].

$$d = \frac{M_1 - M_2}{SD_{pooled}} \tag{2}$$

Where M1 and M2 represent the Means of the experimental and control groups, and $SD_{pooled}$ is the pooled standard deviation, calculated as (Equation 3):

$$SD_{pooled} = \sqrt{\frac{(n_1 - 1) \times SD_1^2 + (n_2 - 1) \times SD_2^2}{n_1 + n_2 - 2}} \tag{3}$$

Where n1 and n2 are the sample sizes and $SD_1$ and $SD_2$ are the standard deviations for each group.

Participants were randomly assigned to the experimental (EX; n = 15) and control (CO; n = 15) groups at a ratio of 1:1 using the Random.org website.

### Participants

Participants were recruited from February 2023 to April 2023. Participant inclusion criteria specified female participants aged between 18 and 40 years, with a body mass index in the normal range (18–24 kg/m$^2$, not overweight) (for both groups), and blind individuals with limited visual acuity of 20/600 and a restricted field of vision of 20°. Participants' overall health was assessed using the General Health Questionnaire before inclusion in the study [24]. Exclusion criteria

comprised individuals with neurological disorders, auditory impairments, history of vestibular system impairment, pathological symptoms, previous fractures or joint-related surgeries, and lower limb diseases. Informed consent was obtained verbally from all participants after providing a detailed explanation of the study's objectives and methods. The consent process included providing participants with a written document in Braille or an audio recording of the consent form to ensure they had full understanding of the study before consenting. The research protocols were approved by the Ethics Committee of Kerman University of Medical Sciences in Iran (IR.KMU.REC.1395.598).

## Setup

Demographic information encompassing age, educational background, state of residence, employment status and marital status was collected from each participant through structured interviews and questionnaires. Subsequent to the demographic data collection, participants were individually acquainted with the experimental tests and procedures. This introductory session aimed to ensure participants comprehended the study's purpose and were familiar with the equipment and protocols involved in data collection, thereby minimizing potential confounding factors like psychological biases or unfamiliarity with testing procedures. During this session, researchers provided detailed explanations of each test and addressed any questions or concerns raised by the participants. Researchers received prior training on effective communication techniques tailored to individuals with visual impairments, ensuring clear, accessible explanations of all procedures. During the session, each test, was explained in detail, with special attention given to how the equipment would be used and the sensations the participants might experience. Researchers used verbal descriptions and tactile demonstrations where appropriate, providing participants with the opportunity to ask questions or express concerns. This approach was designed to enhance participant understanding and comfort with the procedures, reducing any anxiety or confusion about new testing methods. Special attention was given to standardizing the testing procedures across all participants, such as ensuring consistent positioning of equipment, maintaining the same time duration for each test, following identical instructions for every participant, standardized lighting, temperature and noise levels to minimize potential biases or variations in test administration. After the introductory session, participants were scheduled to visit the laboratory of the Faculty of Sports Sciences at Shahid Bahonar University in Kerman for pre-test measurements. Upon arrival at the laboratory, participants underwent further instructions and preparations before proceeding with the pre-test measurements. This included providing a brief review of the testing procedures, explaining the specific tasks they would perform during the pre-test, and ensuring that participants were comfortable with the equipment setup. Participants were also given the opportunity to adjust the equipment if needed and were reminded of any important safety instructions. Additionally, participants were given time to familiarize themselves with the laboratory setting and the testing environment to alleviate any anxiety or discomfort associated with the procedures.

Electromyography (EMG) was used to assess muscle activity during specific tasks. Surface bipolar electrode pairs of the CDE model (OT Bioelettronica, Italy) were utilized for registering muscular activity. These electrodes, consisting of disposable silver/silver chloride (AgCl-Ag), were placed on the skin according to the recommendations outlined in the Surface Electromyography for the Non-Invasive Assessment of Muscles (SENIAM) guidelines [25]. The placement of electrodes on the skin was critical for accurately capturing the electrical signals generated by the underlying muscles. Careful attention was paid to positioning the electrodes over the target muscles of interest, including the tibialis anterior (TA), gastrocnemius medialis (GM), rectus femoris (RF), and biceps femoris (BF). The center-to-center distance between electrode pairs was standardized to two centimeters to optimize signal quality and minimize cross-talk between adjacent muscles. EMG data from the selected muscles were recorded using a 16-channel EMG amplifier (EMG-USB2+, OT-Bioelettronica, Torino, Italy) with a high sampling frequency of 2048 Hz. To ensure a high signal-to-noise ratio, disposable silver/silver chloride electrodes were chosen for their excellent conductivity and biocompatibility with the skin [26]. The ground electrode was securely attached to the lateral malleolus to serve as a reference point for measuring muscle activity.

In addition to EMG recordings, lower limb muscle strength was assessed using a Lafayette Handheld Dynamometer (HHD) model-47904 (Lafayette Instrument Company, Lafayette IN, USA). The HHD is a reliable and valid instrument for

quantifying muscle strength in various populations, with reported intra-rater reliability ranging from 0.93 to 0.98 [27]. To ensure the stability and accuracy of strength measurements, precautions were taken to prevent movement or displacement of the HHD during testing. A strap was used to secure the device firmly in place, minimizing the risk of slippage or misalignment during muscle strength assessments.

## Outcomes

**EMG data analysis.** Following data collection, EMG signals were processed and analyzed to quantify muscle activity during specific tasks or movements. The maximum voluntary isometric contraction (MVIC) method was employed to normalize raw EMG signals, which involved dividing the EMG signals by the corresponding values obtained during maximum contraction of the target muscles. To obtain accurate measurements, participants performed two tests of maximum voluntary isometric contractions for each muscle, with a 2-minute rest period between repetitions. EMG activity was recorded for a duration of 7 seconds, with a specific focus on the middle 3 seconds to capture the peak muscle activity [28]. The highest value obtained from the two tests was selected for further analysis to ensure consistency and reliability of the data. The filtered signals were then processed using root mean square (RMS) averaging over 25-millisecond segments to obtain a representative measure of muscle activity. Muscle activity was quantified by dividing the RMS value by the MVIC and multiplying the result by 100 to express the activity as a percentage of maximum voluntary contraction [25]. During the treadmill movement, EMG activity was recorded in both the anterior-posterior (front-to-back) and posterior-anterior (back-to-front) directions. The treadmill speed was set to 1.1 m/s for the main testing phase, based on pilot testing involving three subjects, which ensured an appropriate speed range (0.8 m/s to 1.2 m/s). Participants stood barefoot on the treadmill (h/p/cosmos-mercury COS 10198 model, Germany), with their hands crossed on their chest and the right side of their body facing the cameras. They were instructed to position their feet hip-width apart, allowing them to find their natural and comfortable stance. Once positioned, participants faced forward and were then randomly directed to turn either towards or away from the treadmill display screen, inducing disturbances to simulate real-life scenarios and minimize order effects. The participants were asked to manage the disturbances without stepping, and if their feet shifted, the movement was repeated. The safety of the participants was ensured by a safety belt suspended from the ceiling at the center of the treadmill, which did not unload body weight. This setup allowed for a comprehensive assessment of muscle recruitment patterns and biomechanical demands.

## Muscle strength measurement

Muscle strength was quantified using the Lafayette Handheld Dynamometer, with specific protocols and procedures tailored to each muscle group of interest (Table 1). Participants were positioned according to standardized guidelines, and the dynamometer was carefully aligned to ensure accurate force measuremenst. Muscle strength was assessed for both

**Table 1. Measuring muscle strength [49].**

| Muscles | Test position | Dynamometer Placement Location |
|---|---|---|
| Ankle dorsiflexors | Sitting position with a 90-degree hip angle and open knee | Lower edge of the HHD sensor is placed above the head of the first metatarsal bone (plantar surface) |
| Ankle Plantar flexors | Sitting position with a 90-degree hip angle and open knee | Lower edge of the HHD sensor is placed above the head of the first metatarsal bone (sole surface) |
| Knee extensors | Sitting position with a 90-degree hip and knee angle | Lower edge of the HHD sensor is placed above the anterior tuberosity (anterior surface of the shin) |
| Knee flexors | Sitting position with a 90-degree hip and knee angle | Lower edge of the HHD sensor is placed above the medial condyle (posterior surface of the shin) |

agonist and antagonist muscle groups to provide a comprehensive evaluation of muscular function and balance around the joints of interest.

## Intervention

After the pre-test, participants in the exercise group embarked on an 8-week aquatic exercise program led by an experienced instructor with seven years of expertise in aquatic exercise. Given the visual impairment of the participants, the instructor was trained in effective communication techniques tailored to individuals with visual impairments. Verbal cues were used extensively throughout the sessions to explain movements and provide guidance on proper form. Additionally, tactile demonstrations and clear, consistent descriptions of the exercises were employed to ensure that all participants fully understood the movements and felt confident in performing them. The instructor maintained an open line of communication, regularly checking in with participants to address any concerns or discomforts and ensuring that participants felt supported throughout the program. The intervention program included three weekly sessions, each lasting around 60 minutes, starting with a 5-minute warm-up and concluding with a 5-minute cool-down. The main 50-minute portion focused on balance exercises in water. The exercises employed in this study (detailed in Table 2) have been widely used in previous research and involved activities such as heel and toe walking, standing on patterned balance boards, using wobble boards and Bosu balls for balance, walking in patterned sandals, and navigating foam or noodle obstacles [29–32]. These were chosen to improve proprioceptive and somatosensory systems. To ensure the appropriateness and effectiveness of the exercises, they were reviewed and validated by a panel of experts in sports science. The experts assessed the exercises based on their relevance to the study population, and their feasibility for implementation in water-based settings. This consultation ensured that the protocol aligned with best practices and evidence-based methodologies. Additionally, the exercise difficulty gradually increased over the 8 weeks by modifying hand positions (e.g., opening and closing hands to maintain balance) or by intensifying the activities (e.g., increasing walking distances or durations, standing on tiptoes or heels, incorporating more balance boards, extending strides, or performing exercises with eyes closed).

**Table 2. Details of aquatic exercise protocol.**

| | Exercises |
|---|---|
| **Warm-up (5 minutes)** | - Increasing walking speed<br>- Stretching exercises for major muscle groups |
| **Main exercises (50 minutes)** | - Walking forward with straight knees (3m)<br>- Walking backward with long strides (3m)<br>- Walking laterally with long strides (3m)<br>- Walking on tiptoes (3m)<br>- Walking with trunk rotation (3m)<br>- Walking with consecutive single-leg stops (3m)<br>- Flexion and extension of the hip in a single-leg - position with straight knees (3 s × 12 rep)<br>- Flexion and extension of shoulder joints in a half-squat position (3 s × 12 rep)<br>- Horizontal abduction and adduction of the shoulder joints in a half-squat position (3 s × 12 rep)<br>- Walking on toes and heels (3 m)<br>- Performing simultaneous extension of the hip and knee with plantar flexion of the ankle, followed by thigh and knee flexion with dorsiflexion of the ankle (3 s × 12 rep)<br>- Standing on a patterned balance board (2 m)<br>- Walking with patterned sandals (2 m)<br>- Walking on a patterned floor (2 m)<br>- Standing on a balance board (2 m) |
| **Cool-down (5 minutes)** | - Walking<br>- Stretching exercises for upper and lower body muscles |

Note: m = minute, s = set, rep = repetition

Meanwhile, the control group did not engage in specific sports activities, focusing instead on their usual daily routines. Initially, no formal record or questionnaire captured their daily activities. However, to reduce potential biases, a researcher conducted random weekly check-ins with control group members to monitor their activities and general well-being throughout the study. At the study's conclusion, all participants were evaluated with post-tests, and the findings were recorded. The study ensured consistent conditions for all participants, including standardized lighting, temperature, noise levels, and testing procedures.

## Statistical analysis

The data collected for the research variables underwent statistical analysis using SPSS 26.0 for Windows (SPSS Inc., Chicago, IL, USA). The Shapiro-Wilk test was used to confirm the normality of the distribution. A mixed-design analysis of variance (ANOVA) was conducted on all variables, with the time factor (pre-test and post-test) considered within subjects, and the group factor (EX and Co) as a between-subjects factor. Baseline demographics, muscle activation and strength values were compared between the two groups using independent t-tests, and no significant differences were found, ensuring that any group differences at post-test were not confounded by baseline disparities. If a significant interaction was found between factors, post hoc paired t-tests with Bonferroni correction for multiple comparisons were applied. The effect sizes (ES) for all parameters were calculated using partial $\eta^2$. In this context, a partial $\eta^2$ value of 0.02, 0.13, and 0.26 represented small, medium, and large effect sizes, respectively [33]. Moreover, in conjunction with determining the statistical significance of differences between pre-test and post-test measurements, Cohen's d was calculated to assess the practical significance of these changes. For interpretation, a Cohen's d of 0.2, 0.5, and 0.8 or above represent small, medium, and large effect sizes, respectively [33]. The significance level was set at $p < 0.05$.

## Results

### Participant flow and demographic characteristics

The flow of participants through the trial is presented in Fig 1. Initially, 54 participants were screened for eligibility. Among them, 24 were excluded for not meeting the inclusion criteria. During the 8-week intervention period, no participants withdrew from the study. Finally, the results of 15 participants in the CO group and 15 participants in the EX group were included in the final analysis. Table 3 presents the demographic characteristics of the participants. The independent t-test results showed that there were no significant differences in demographics or baseline muscle activation and strength between the two groups, ensuring comparability at baseline. Fig 1 near here.

### EMG data

**Muscle activation (MVIC%).** The mixed ANOVA results in Table 4 showed a significant interaction between group and time ($p < 0.05$), as well as main effects for time ($p < 0.05$) and group ($p < 0.05$) for MVIC% in both anterior-posterior and posterior-anterior directions. However, in the posterior-anterior direction, the group effect and the interaction between group and time for the RF muscle were not significant ($p > 0.05$). Post hoc analyses, adjusted for multiple comparisons using the Bonferroni correction, revealed significant improvements in MVIC% for the TA (t = 4.61, p = 0.001, d = 1.25), RF (t = 2.73, p = 0.01, d = 0.74), and BF (t = 3.08, p = 0.004, d = 0.87) muscles in the anterior-posterior direction. However, no significant differences were found for the GM muscle (t = 1.94, p = 0.06, d = 0.52). In the posterior-anterior direction, significant differences were observed for the GM (t = 5.05, p = 0.001, d = 1.15) and BF muscles (t = 4.49, p = 0.001, d = 1.02), while differences for the TA muscle were not significant (t = 1.47, p = 0.15, d = 0.38).

### Muscle activation timing (onset time)

Significant interactions and main effects ($p < 0.05$) were observed for onset time in both anterior-posterior and posterior-anterior directions. Post hoc tests, corrected for multiple comparisons using the Bonferroni method, indicated

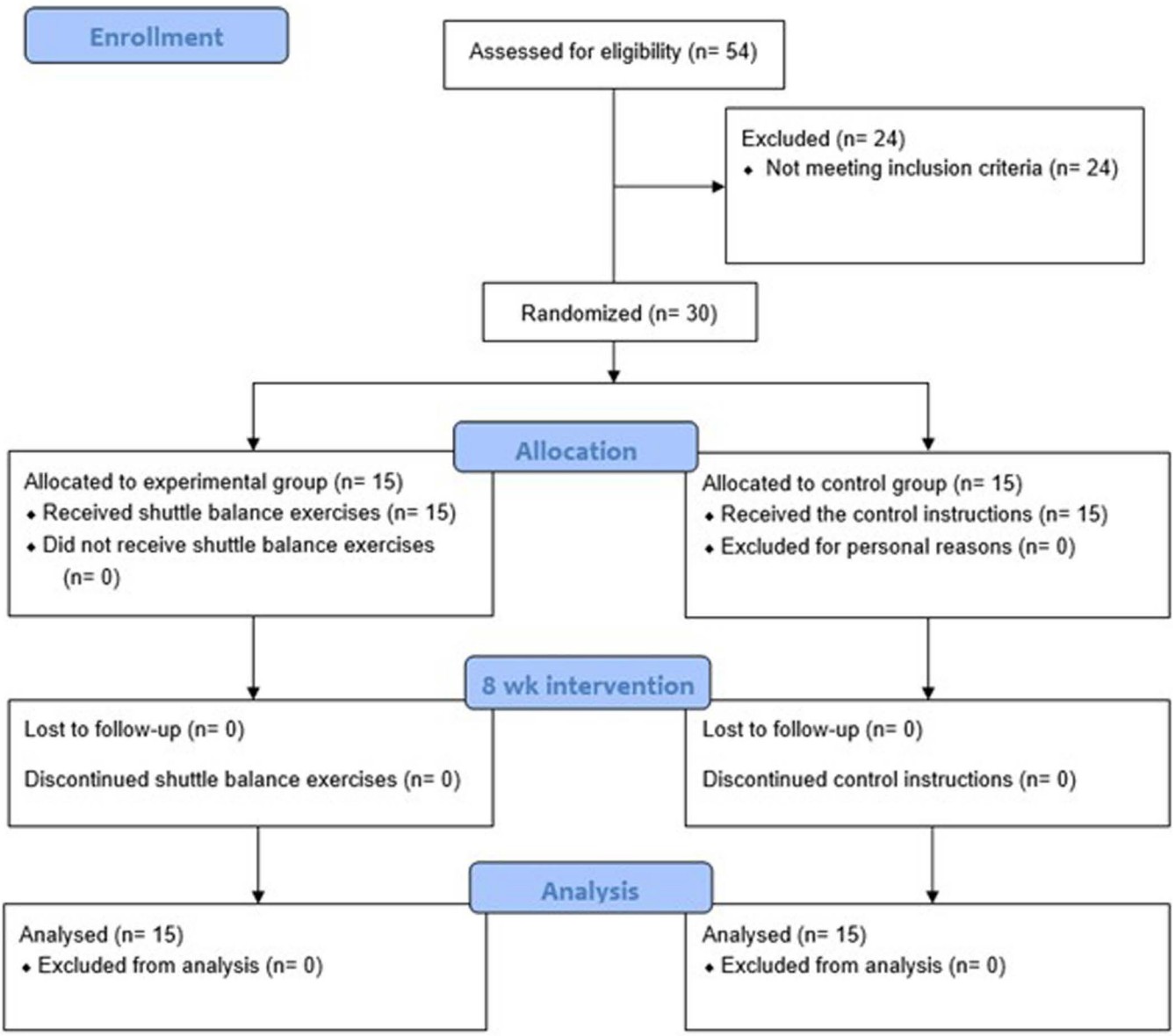

**Fig 1. SPIRIT Schedule of Enrollment.**

significant pre-test to post-test improvements for all muscles except for the GM muscle in the anterior-posterior direction. Significant differences were observed for the TA (t=4.17, p=0.001, d=0.93), GM (t=4.47, p=0.001, d=0.98), RF (t=4.20, p=0.001, d=0.94), and BF muscles (t=4.30, p=0.001, d=1.01). Similarly, in the posterior-anterior direction, significant improvements were observed in the TA muscle (t=4.27, p=0.001, d=1.05), GM muscle (t=3.89, p=0.001, d=0.98), RF muscle (t=3.33, p=0.002, d=0.85), and BF muscle (t=9.53, p=0.001, d=2.34).

## Muscle strength

Table 4 presents the pre-test and post-test muscle strength data. The mixed ANOVA results revealed a significant interaction between group and time for ankle dorsiflexors, as well as a significant time effect and interaction between group

**Table 3. Demographic characteristics of studied groups Reported as mean (SD).**

|  | EX | CO | P |
|---|---|---|---|
| Age (year) | 28.93 (2.25) | 29.13 (2.23) | 0.98 |
| Height (m) | 1.58 (5.19) | 1.57 (7.48) | 0.73 |
| Mass (kg) | 53.66 (4.48) | 56.13 (6.82) | 0.25 |
| BMI (kg/m²) | 21.47 (1.77) | 22.61 (1.50) | 0.06 |

Note: EX = experimental, CO = control

Significant level set as $p < 0.05$.

and time for knee extensors ($p < 0.05$). Moreover, no significant changes ($p > 0.05$) were found for ankle plantar flexors, knee flexors, or ankle dorsiflexors. Group effects indicated significant differences in ankle plantar flexors and dorsiflexors ($p < 0.05$), but not in knee extensors or flexors ($p > 0.05$). Post hoc analyses, adjusted for multiple comparisons using the Bonferroni correction, showed significant improvements in knee extensors ($t = -2.93$, $p = 0.007$, $d = 0.36$), but no significant changes were found for ankle dorsiflexors ($t = -1.26$, $p = 0.21$, $d = 0.32$).

## Understanding Interactions

The significant interaction effects observed in both EMG data (MVIC% and onset time) and muscle strength indicate that the changes over time varied between the EX and CO groups. Specifically, the intervention led to notable improvements in muscle activation and strength metrics in the EX group compared to the CO group, emphasizing the differential impact of the aquatic exercises. The nonsignificant interaction effects for certain variables, such as ankle dorsiflexors, suggest that while improvements were observed, they were not significantly different between groups over time.

## Discussion

The study examined the effects of aquatic exercises on EMG activation and muscle strength in women with visual impairments, revealing significant improvements in muscle activation (MVIC%) and onset timing for TA, RF, GM, and BF, as well as increased strength in ankle dorsiflexors and knee extensors. These findings suggest enhanced neuromuscular function, with potential benefits for stability and fall prevention.

Enhanced neuromuscular control allows individuals with visual impairments to better anticipate and respond to perturbations, reducing fall risks. The observed improvements in EMG activation align with previous research demonstrating the benefits of aquatic exercises on muscle function and coordination across different populations [34,35]. Aquatic exercises have been shown to increase muscle strength, proprioceptive feedback, and neuromuscular coordination in groups such as stroke patients and individuals with balance impairments [17,36,37]. The buoyant environment of water reduces the fear of falling, promotes movement freedom, and enhances proprioception, which is essential for postural stability [32,38]. The adaptations observed in muscle spindles and Golgi tendon organs indicate that aquatic exercises contribute to heightened neuromuscular control [39,40]. Increased muscle spindle sensitivity enhances afferent input to the central nervous system, improving balance and stability. These findings suggest that aquatic exercises help individuals with visual impairments optimize motor unit activation and muscle recruitment, thereby strengthening the lower limbs and reducing fall risks [41]. Aquatic exercises influence muscle activation patterns during perturbation responses. For example, posterior-anterior perturbations primarily activate the gastrocnemius, while anterior-posterior perturbations engage the quadriceps to prevent forward falls. Increased activation of these muscles following aquatic training helps regulate lower limb movements, improving dynamic stability [34]. The inhibitory effect of Golgi tendon receptors on force production is also reduced, allowing for greater muscle engagement and force output, which further contributes to improved functional stability [39,42].

**Table 4. The data regarding the examined groups during both the pre-test and post-test. Mean (SD).**

| | | EX | | CO | | Time Effect | Group Effect | Group × Time Interaction |
|---|---|---|---|---|---|---|---|---|
| | | Pre- test | Post-test | Pre- test | Post-test | | | |
| **EMG data (MVIC%)** | | | | | | | | |
| Anterior-Posterior | TA | 12.52 (1.02) | 16.95 (0.22) | 12.91 (0.72) | 12.67 (0.75) | F = 233.22 P = 0.001 ES = 0.89 | F = 110.55 P = 0.001 ES = 0.79 | F = 289.58 P = 0.001 ES = 0.91 |
| | GM | 10.56 (0.45) | 11.49 (0.82) | 10.42 (0.60) | 10.33 (0.86) | F = 4.52 P = 0.04 ES = 0.13 | F = 14.88 P = 0.001 ES = 0.34 | F = 6.66 P = 0.01 ES = 0.19 |
| | RF | 14.80 (0.56) | 16.40 (0.95) | 14.88 (0.41) | 14.63 (0.57) | F = 14.08 P = 0.001 ES = 0.33 | F = 27.42 P = 0.001 ES = 0.49 | F = 26.72 P = 0.001 ES = 0.48 |
| | BF | 7.67 (0.60) | 9.48 (0.56) | 7.98 (0.72) | 7.75 (0.67) | F = 20.45 P = 0.001 ES = 0.42 | F = 20.03 P = 0.001 ES = 0.41 | F = 34.40 P = 0.001 ES = 0.55 |
| Posterior-Anterior | TA | 13.76 (1.17) | 15.31 (0.92) | 13.62 (1.11) | 13.09 (1.08) | F = 3.07 P = 0.09 ES = 0.09 | F = 19.15 P = 0.001 ES = 0.40 | F = 12.87 P = 0.001 ES = 0.31 |
| | GM | 14.81 (0.64) | 18.70 (0.84) | 18.77 (0.70) | 14.46 (1.06) | F = 105.66 P = 0.001 ES = 0.79 | F = 109.63 P = 0.001 ES = 0.79 | F = 92.09 P = 0.001 ES = 0.76 |
| | RF | 9.69 (1.03) | 10.65 (1.45) | 9.55 (0.78) | 9.76 (0.94) | F = 6.70 P = 0.01 ES = 0.19 | F = 2.54 P = 0.12 ES = 0.08 | F = 2.78 P = 0.10 ES = 0.09 |
| | BF | 15.62 (0.86) | 17.61 (1.09) | 15.49 (0.74) | 15.54 (0.74) | F = 51.11 P = 0.001 ES = 0.64 | F = 14.90 P = 0.001 ES = 0.34 | F = 45.49 P = 0.001 ES = 0.61 |
| **Onset time (ms)** | | | | | | | | |
| Anterior-Posterior | TA | 275.72 (21.53) | 192.86 (12.54) | 283.58 (19.40) | 290.59 (21.45) | F = 109.48 P = 0.001 ES = 0.79 | F = 67.21 P = 0.001 ES = 0.70 | F = 153.78 P = 0.001 ES = 0.84 |
| | GM | 295.72 (20.59) | 200.94 (14.17) | 303.57 (24.50) | 310.59 (21.35) | F = 269.41 P = 0.001 ES = 0.90 | F = 70.60 P = 0.001 ES = 0.71 | F = 362.52 P = 0.001 ES = 0.92 |
| | RF | 294.37 (17.59) | 194.93 (21.18) | 311.54(19.29) | 319.78 (15.23) | F = 114.68 P = 0.001 ES = 0.80 | F = 264.96 P = 0.001 ES = 0.90 | F = 159.86 P = 0.001 ES = 0.85 |
| | BF | 287.78 (16.42) | 201.94 (14.17) | 295.88 (19.19) | 301.96 (12.32) | F = 206.11 P = 0.001 ES = 0.88 | F = 55.84 P = 0.001 ES = 0.66 | F = 288.41 P = 0.001 ES = 0.91 |
| Posterior-Anterior | TA | 307.20 (13.35) | 207.42 (12.04) | 314.26 (23.46) | 320.89 (15.87) | F = 97.89 P = 0.001 ES = 0.77 | F = 236.43 P = 0.001 ES = 0.89 | F = 127.72 P = 0.001 ES = 0.82 |
| | GM | 269.66 (10.90) | 187.97 (8.70) | 272.32 (18.54) | 282.53 (10.90) | F = 109.04 P = 0.001 ES = 0.79 | F = 231.78 P = 0.001 ES = 0.89 | F = 180.24 P = 0.001 ES = 0.86 |
| | RF | 280.76 (11.79) | 206.67 (27.84) | 284.56 (20.88) | 294.96 (12.79) | F = 33.30 P = 0.001 ES = 0.54 | F = 105.78 P = 0.001 ES = 0.79 | F = 58.64 P = 0.001 ES = 0.67 |
| | BF | 343.17 (14.94) | 208.03 (11.24) | 349.09 (13.13) | 306.58 (13.53) | F = 594.02 P = 0.001 ES = 0.95 | F = 266.67 P = 0.001 ES = 0.90 | F = 161.47 P = 0.001 ES = 0.85 |

*(Continued)*

**Table 4.** (Continued)

| | EX | | CO | | Time Effect | Group Effect | Group × Time Interaction |
|---|---|---|---|---|---|---|---|
| | Pre- test | Post-test | Pre- test | Post-test | | | |
| **Strength (N/kg)** | | | | | | | |
| Ankle dorsiflexors | 17.55 (4.27) | 20.71 (1.80) | 17.07 (4.37) | 16.29 (1.97) | F = 1.83 P = 0.18 ES = 0.06 | F = 8.32 P = 0.007 ES = 0.22 | F = 5.04 P = 0.03 ES = 0.15 |
| Ankle plantar flexors | 21.67 (5.60) | 25.07 (3.04) | 20.67 (5.61) | 20.83 (5.76) | F = 1.93 P = 0.17 ES = 0.06 | F = 5.54 P = 0.02 ES = 0.16 | F = 2.47 P = 0.12 ES = 0.08 |
| Knee extensors | 25.73 (2.68) | 27.96 (3.53) | 25.25 (2.82) | 25.24 (2.76) | F = 11.81 P = 0.002 ES = 0.29 | F = 2.39 P = 0.13 ES = 0.07 | F = 11.89 P = 0.002 ES = 0.29 |
| Knee flexors | 21.88 (2.61) | 22.62 (2.30) | 21.14 (2.68) | 21.13 (2.60) | F = 2.75 P = 0.10 ES = 0.08 | F = 0.06 P = 0.80 ES = 0.002 | F = 2.87 P = 0.10 ES = 0.09 |

Note: EMG = electromyography, MVIC = maximum voluntary isometric contraction, ES = effect size, CI = confidence interval, TA = tibialis anterior, GM = gastrocnemius medialis, RF = rectus femoris, BF = biceps femoris, EX = experimental, CO = control

Significant level set as $p < 0.05$.

The improvements in muscle strength and faster onset times observed in this study have clinical significance for individuals with visual impairments. Faster onset times indicate more efficient muscle recruitment, which is crucial for postural stability and fall prevention. The study's findings are consistent with research demonstrating increased thigh abductor activation and simultaneous contraction of stabilizing muscles following aquatic exercises [43,44]. Such adaptations enhance functional joint stability, optimize neuromuscular efficiency, and improve dynamic balance [35,38]. The resistance properties of water provide a unique training stimulus that enhances muscle strength while minimizing joint stress [45]. Participants in the exercise group showed greater improvements in lower limb strength than the control group, particularly in ankle dorsiflexors, plantar flexors, and knee extensors, consistent with prior studies on aquatic rehabilitation [46,47]. Increased proprioceptive and somatosensory demands during aquatic training likely contributed to these strength gains [48]. Previous studies have also demonstrated increased lower limb strength following aquatic exercise programs in older adults and individuals with osteoarthritis [18]. The unique properties of water, including buoyancy and hydrodynamic resistance, create an ideal environment for rehabilitation, particularly for individuals with physical impairments. Water-based training allows for greater movement control, improved balance, and enhanced neuromuscular function while reducing impact forces on joints. As a result, aquatic exercises serve as an effective strategy for improving functional ability and reducing fall risks in individuals with visual impairments.

In summary, the concurrent improvements in muscle activation, strength, and neuromuscular coordination underscore the holistic benefits of aquatic exercises for individuals with visual impairments. By targeting sensorimotor deficits and enhancing proprioception, aquatic exercises offer a promising approach to improving balance, reducing fall risks, and promoting overall functional well-being.

## Limitations

It is essential to acknowledge certain limitations of the study. While the results demonstrated significant improvements in muscle activation and strength, the study did not directly investigate the effects of these improvements on postural control or functional outcomes, such as fall prevention. Additionally, the study's relatively short duration limits our understanding

of the long-term sustainability of the observed improvements. Future research should explore the relationship between muscle activity, strength, and postural control to provide a more comprehensive understanding of the benefits of aquatic exercises. Moreover, more definitive studies with larger sample sizes and extended follow-up periods are needed to assess the durability and broader impact of these interventions over time.

## Conclusion

This study evaluates an aquatic exercise protocol for women who are blind, finding it improves EMG activation, reduces onset time, and increases muscle strength in the lower limbs. These improvements address motor skill deficits related to visual impairment and enhanced muscular coordination and strength. Incorporating aquatic exercises into rehabilitation for individuals with visual disabilities could improve functional capacity and quality of life. Research advocates for aquatic exercises for people with visual impairments for their potential benefit in reducing falls and improving well-being, suggesting further research to develop personalized evidence-based interventions.

## Supporting information

**S1 File. CONSORT-2010-Checklist.**
(DOCX)

**S2 File. Research Protocol – English.**
(DOCX)

**S3 File. Research Protocol - Persian.**
(DOCX)

## Acknowledgments

We extend our heartfelt appreciation to all the participants who generously dedicated their time and efforts to contribute to this study.

## Author contributions

**Conceptualization:** Asma Salari, Mansour Sahebozamani.

**Data curation:** Asma Salari, Mohammad Alimoradi, Mojtaba Iranmanesh.

**Formal analysis:** Abdolhamid Daneshjoo, Mohammad Alimoradi.

**Investigation:** Asma Salari, Mohammad Alimoradi, Mojtaba Iranmanesh, Guillermo Mendez-Rebolledo.

**Methodology:** Asma Salari, Mansour Sahebozamani.

**Project administration:** Mansour Sahebozamani.

**Resources:** Abdolhamid Daneshjoo.

**Software:** Mohammad Alimoradi.

**Supervision:** Mansour Sahebozamani, Abdolhamid Daneshjoo, Nicola Relph.

**Validation:** Mansour Sahebozamani, Abdolhamid Daneshjoo.

**Visualization:** Asma Salari, Mansour Sahebozamani, Nicola Relph.

**Writing – original draft:** Asma Salari, Mojtaba Iranmanesh.

**Writing – review & editing:** Abdolhamid Daneshjoo, Nicola Relph, Guillermo Mendez-Rebolledo.

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
