## [Decision Letter · Decision Letter 0]

4 Mar 2025

PONE-D-25-04750Effects of an aquatic protocol on electromyography activation and strength of lower limb muscles in blind women. A randomized controlled trialPLOS ONE

Dear Dr. Relph,

Thank you for submitting your manuscript to PLOS ONE. After careful consideration, we feel that it has merit but does not fully meet PLOS ONE’s publication criteria as it currently stands. Therefore, we invite you to submit a revised version of the manuscript that addresses the points raised during the review process.

We look forward to receiving your revised manuscript.

Kind regards,

Luciana Labanca

Academic Editor

PLOS ONE

Reviewers' comments:

Reviewer's Responses to Questions

**Comments to the Author**

1. Is the manuscript technically sound, and do the data support the conclusions?

Reviewer #1: Partly

Reviewer #2: Partly

2. Has the statistical analysis been performed appropriately and rigorously? 

Reviewer #1: No

Reviewer #2: Yes

3. Have the authors made all data underlying the findings in their manuscript fully available?

Reviewer #1: Yes

Reviewer #2: Yes

4. Is the manuscript presented in an intelligible fashion and written in standard English?

Reviewer #1: Yes

Reviewer #2: Yes

5. Review Comments to the Author

Reviewer #1: This, at best, appears to be a pilot effort. There are many analyses in this paper having only 30 observations. The design is inadequately justified. A sample size of 15 per group appears a bit limiting for confident conclusions. The sample size formula (equation 1)was presented but with little detail. The parameters used in the formula ,e.g. means and variance, should be specified. How was the effect size of 0.46 derived from actual past data?

Table 4 appears with many p-values or multiple comparisons. Were the p-values adjusted for Bonferroni correction as stated in the protocol supplement? This is not obvious in the manuscript.

The authors state that more definitive studies with more time sustainable effects may be needed. They should also include this in the limitations and add that the studies should be larger.

Reviewer #2: The current manuscript presents the results on the effects of an aquatic exercise routine on the EMG findings and muscle strength in blind women.

Were the baseline EMG indices and muscle strength comparable/randomized between the two groups?

The introduction is well prepared but also very redundant. Please remove the unnecessary repetition as it takes away from clarity and conciseness of the text. As an example the last paragraph of the introduction is mostly repeated sentences. The problem of redundancy also affects the discussion. I encourage the authors to make the text more concise and shorten and remove unnecessary content that doesn’t add value to the text.

The explanation of the power analysis formula is out of scope for this paper. Mentioning the α and β values, the effect size and the resulting sample size is sufficient.

For start and end of the subject recruitment, mentioning month and year is more than enough.

6. PLOS authors have the option to publish the peer review history of their article (what does this mean? ). If published, this will include your full peer review and any attached files.

**Do you want your identity to be public for this peer review?** For information about this choice, including consent withdrawal, please see our Privacy Policy .

Reviewer #1: No

Reviewer #2: No

---

## [Author Response · Author response to Decision Letter 1]

11 Mar 2025

Dear Reviewers,

Thank you for your valuable feedback on our manuscript titled " Effects of an aquatic protocol on electromyography activation and strength of lower limb muscles in blind women. A randomized controlled trial " We have carefully addressed your comments and made significant revisions to improve the quality of our work. Below, you will find our detailed responses to each of your suggestions. We appreciate your insights and look forward to your thoughts on the revised manuscript.

Best regards,

# Reviewer 1

1. This, at best, appears to be a pilot effort. There are many analyses in this paper having only 30 observations. The design is inadequately justified. A sample size of 15 per group appears a bit limiting for confident conclusions. The sample size formula (equation 1)was presented but with little detail. The parameters used in the formula ,e.g. means and variance, should be specified. How was the effect size of 0.46 derived from actual past data?

Response: Thank you for your feedback. We have revised the manuscript to provide a clearer justification for the sample size and effect size calculation. The sample size was determined using a 95% confidence level and 80% statistical power, with an effect size of 0.46 based on prior research on aquatic exercise intervention for visually impaired individuals (Salari et al. 2024). This calculation indicated a minimum of 12 participants per group, but to account for potential dropouts, we increased it to 15 per group (n = 30 total) to ensure statistical robustness. The effect size was calculated using Cohen’s d, considering the means and pooled standard deviation of the two groups, following established statistical principles. Additionally, we have strengthened our justification for the study design by emphasizing that this is a single-blind randomized controlled trial (RCT). Randomization was performed using Random.org, ensuring balanced group allocation and minimizing selection bias. We hope this provides more justification for the study design.

Salari, A., Sahebozamani, M., Daneshjoo, A., Alimoradi, M., Iranmanesh, M., & Mendez-Rebolledo, G. (2024). Effects of 8 weeks aquatic exercises on balance recovery strategies and center of pressure sways in blind women: A randomized controlled trial. British Journal of Visual Impairment, 0(0), 02646196241281254. https://doi.org/10.1177/02646196241281254

2. Table 4 appears with many p-values or multiple comparisons. Were the p-values adjusted for Bonferroni correction as stated in the protocol supplement? This is not obvious in the manuscript.

Response: Thank you for your valuable feedback. We have revised the manuscript to clarify the use of the Bonferroni correction in the post hoc analyses. We hope the revised version meets your expectations.

3. The authors state that more definitive studies with more time sustainable effects may be needed. They should also include this in the limitations and add that the studies should be larger.

Response: Thank you for your feedback. In response to your suggestion, we have updated the limitations section to highlight the need for larger studies with extended follow-up periods to assess the sustainability of the observed improvements. Additionally, we have emphasized the importance of investigating the relationship between muscle activation, strength, postural control, and functional outcomes like fall prevention in future research. We hope these revisions address your concerns and improve the manuscript.

# Reviewer 2

1. Were the baseline EMG indices and muscle strength comparable/randomized between the two groups?

Response: Thank you for your insightful comment. We can confirm that baseline EMG indices and muscle strength were comparable between the two groups. Randomization was conducted to ensure that any potential differences between groups at baseline were minimized. We have clarified this in the revised manuscript to provide more transparency on the randomization process and the baseline measures.

2. The introduction is well prepared but also very redundant. Please remove the unnecessary repetition as it takes away from clarity and conciseness of the text. As an example the last paragraph of the introduction is mostly repeated sentences. The problem of redundancy also affects the discussion. I encourage the authors to make the text more concise and shorten and remove unnecessary content that doesn’t add value to the text.

Response: Thank you for your valuable feedback. We have carefully revised the introduction and discussion sections to remove redundancy and improve clarity and conciseness. Specifically, we have shortened the last paragraph of the introduction and streamlined the discussion to eliminate unnecessary repetition while maintaining the key messages of our study.

3. The explanation of the power analysis formula is out of scope for this paper. Mentioning the α and β values, the effect size and the resulting sample size is sufficient.

Response: Thank you for your valuable feedback. In response to Reviewer 1’s comments, we revised the sample size and power analysis section for greater clarity. We expanded on the statistical parameters used in the sample size calculation, specifically the effect size (0.46), confidence level (α = 0.05), and statistical power (β = 0.80). The effect size of 0.46 was derived from previous studies, such as Salari et al. (2024), which focused on aquatic exercise interventions in visually impaired individuals, ensuring its relevance to our study. Following Reviewer 1’s suggestion, we simplified the explanation of the power analysis, retaining only the key details necessary for understanding the sample size determination. These adjustments aim to provide transparency and improve the rigor of our study design while keeping the explanation focused and appropriate for the paper’s scope.

Salari, A., Sahebozamani, M., Daneshjoo, A., Alimoradi, M., Iranmanesh, M., & Mendez-Rebolledo, G. (2024). Effects of 8 weeks aquatic exercises on balance recovery strategies and center of pressure sways in blind women: A randomized controlled trial. British Journal of Visual Impairment, 0(0), 02646196241281254. https://doi.org/10.1177/02646196241281254

4. For start and end of the subject recruitment, mentioning month and year is more than enough.

Response: Thank you for your suggestion. We have revised the manuscript to mention only the month and year for the start and end of subject recruitment.

---

## [Decision Letter · Decision Letter 1]

21 Mar 2025

Effects of an aquatic protocol on electromyography activation and strength of lower limb muscles in blind women. A randomized controlled trial

PONE-D-25-04750R1

Dear Dr. Relph,

We’re pleased to inform you that your manuscript has been judged scientifically suitable for publication and will be formally accepted for publication once it meets all outstanding technical requirements.

Kind regards,

Luciana Labanca

Academic Editor

PLOS ONE

Additional Editor Comments (optional):

Reviewers' comments:

Reviewer's Responses to Questions

**Comments to the Author**

1. If the authors have adequately addressed your comments raised in a previous round of review and you feel that this manuscript is now acceptable for publication, you may indicate that here to bypass the “Comments to the Author” section, enter your conflict of interest statement in the “Confidential to Editor” section, and submit your "Accept" recommendation.

Reviewer #1: All comments have been addressed

2. Is the manuscript technically sound, and do the data support the conclusions?

Reviewer #1: (No Response)

3. Has the statistical analysis been performed appropriately and rigorously? 

Reviewer #1: (No Response)

4. Have the authors made all data underlying the findings in their manuscript fully available?

Reviewer #1: (No Response)

5. Is the manuscript presented in an intelligible fashion and written in standard English?

Reviewer #1: (No Response)

6. Review Comments to the Author

Reviewer #1: (No Response)

7. PLOS authors have the option to publish the peer review history of their article (what does this mean? ). If published, this will include your full peer review and any attached files.

**Do you want your identity to be public for this peer review?** For information about this choice, including consent withdrawal, please see our Privacy Policy .

Reviewer #1: No

---

## [Editor Report · Acceptance letter]

PONE-D-25-04750R1

PLOS ONE

Dear Dr. Relph,

I'm pleased to inform you that your manuscript has been deemed suitable for publication in PLOS ONE. Congratulations! Your manuscript is now being handed over to our production team.

Kind regards,

on behalf of

Dr. Luciana Labanca

Academic Editor

PLOS ONE